# The fate of the spin polaron in the 1D antiferromagnets

Piotr Wrzosek[1], Adam Kłosiński[1], Yao Wang[2,3], Mona Berciu[4,5], Cliò Efthimia Agrapidis[1]
Krzysztof Wohlfeld[1*]

**1** Institute of Theoretical Physics, Faculty of Physics, University of Warsaw, PL 02093, Poland
**2** Department of Physics and Astronomy, Clemson University, SC 29634, USA
**3** Department of Chemistry, Emory University, Atlanta, GA 30322, USA
**4** Department of Physics and Astronomy, University of British Columbia, V6T 1Z4, Canada
**5** Quantum Matter Institute, University of British Columbia, V6T 1Z4, Canada
\* krzysztof.wohlfeld@fuw.edu.pl

## Abstract

The stability of the spin polaron quasiparticle, well established in studies of a single hole in the 2D antiferromagnets, is investigated in the 1D antiferromagnets using a $t$–$J$ model. We perform an exact slave fermion transformation to the holon-magnon basis, and diagonalize numerically the resulting model in the presence of a single hole. We prove that the spin polaron is stable for any strength of the magnon-magnon interaction *except* for the unique value of the SU(2)-symmetric 1D $t$–$J$ model. Fine-tuning to this unique value is extremely unlikely to occur in *quasi*-1D antiferromagnets, therefore the spin polaron is the stable quasiparticle of real 1D materials. Our results lead to a new interpretation of the ARPES spectra of *quasi*-1D antiferromagnets in the spin polaron language.

# 1   Introduction

A central problem in the study of strongly correlated systems is to understand the differences between quantum many-body systems that have stable low-energy quasiparticles, and those that do not [1–5]. A famous example, which we revisit, relates to expected fundamental differences between the low-energy physics of 1D and 2D antiferromagnets doped with a single hole. The widely accepted paradigm is that in a 2D antiferromagnet, the hole is dressed with collective 2D spin excitations (magnons) and together they form a spin polaron quasiparticle [6–15], whereas in 1D, the spin polaron is unstable to splitting into an elementary 1D spin excitation (spinon) and a spinless hole (holon), a phenomenon called spin-charge separation [16–24].

The paradigmatic explanation for this difference relies on the fact that spinons (magnons) are well-defined collective excitations in 1D (2D) antiferromagnets [1]. Because our goal is to understand the *intrinsic* origin of the different single hole behaviour in 1D and 2D antiferromagnets, we have to use the same language to describe both cases. As the 1D case is always easier to study [25], we choose to recast the 1D problem using the 2D magnon language so that we can answer the question: what is the fate of the spin polaron in the 1D antiferromagnets?

In this paper we answer this question by: (i) developing a novel numerical simulation of the 1D $t$–$J$ model in the magnon-holon basis [8], and (ii) performing a detailed finite size scaling of the quasiparticle properties. We show that the spin polaron quasiparticle is destroyed in the ground state of the 1D antiferromagnet with a single hole *only* when the magnon-magnon interaction are precisely tuned to the unique value dictated by the 1D $t$–$J$ model. For any other value of the magnon-magnon interaction, whether stronger or weaker than this critical value [1], the spin polaron is the stable quasiparticle of the 1D antiferromagnet.

Moreover, we show that the intrinsic, staggered magnetic field present in *quasi*-1D antiferromagnets of real materials [26–28] disrupts this fine balance between the on-site magnon energy and the magnon-magnon interaction of the 1D $t$–$J$ model. This makes the spin polaron quasiparticle stable in the *quasi*-1D cuprates and leads to the interpretation of the ARPES spectra [29] of *quasi*-1D cuprates [18–20, 22, 23] in the spin polaron language. Altogether, these results show an unexpected, impressive robustness of the spin polaron picture in the *quasi*-1D antiferromagnets, proving that the accepted spin-charge separation paradigm is in fact an exception [30–35], not the rule [25]. The obtained results have important consequences reaching beyond condensed matter, *inter alia* in the interpretation of cold atom experiments [13, 36].

The paper is organised as follows. In Sec. 2 we express the 1D $t$–$J$ model in the magnon-holon basis. The obtained in this way fermion-boson model is then generalised by allowing the strength of boson-boson interaction to be tunable—this allows us to study the impact of the magnon-magnon interaction on the properties of the 1D $t$–$J$ model. We solve the problem using exact diagonalisation and show in Sec. 3 how the ground state (3.1) and the excited state (3.2) properties of the single hole in 1D antiferromagnet change once the magnon-magnon interaction is switched off. Next, in Sec. 4 we expand this discussion to the case of varying strength of magnon-magnon interaction and explain that solely its value given by the 1D $t$–$J$ model is critical and leads to suppression of the spin polaron. Finally, in Sec. 5 we show that such a critical value is never reached in realistic materials, such as *quasi*-1D cuprates—hence showing that in this case the spin polaron solution is always stabilised. The paper ends with a short summary section 6 and is supplemented by two appendices, (A and B), which are referred to in the appropriate sections of the main text.

---

[1]Interestingly, this situation is distinct from the one reported in [5], in which interactions *support* the stability of a quasiparticle.

## 2   Model and methods

The Hamiltonian of the standard model of a doped antiferromagnetic chain, the $t$–$J$ model [37], reads,

$$\mathcal{H} = -t \sum_{\langle i,j \rangle, \sigma} \left( \tilde{c}_{i,\sigma}^\dagger \tilde{c}_{j,\sigma} + \text{h.c.} \right) + J \sum_{\langle i,j \rangle} \left( \mathbf{S}_i \cdot \mathbf{S}_j - \frac{1}{4} \tilde{n}_i \tilde{n}_j \right), \tag{1}$$

where $\tilde{c}_{i,\sigma}^\dagger = c_{i,\sigma}^\dagger (1 - n_{i,\bar{\sigma}})$ creates the electron only on unoccupied site, $n_{i,\sigma} = c_{i,\sigma}^\dagger c_{i,\sigma}$ and $\tilde{n}_i = \sum_\sigma \tilde{c}_{i,\sigma}^\dagger \tilde{c}_{i,\sigma}$. Moreover, $\mathbf{S}_i$ are spin-1/2 Heisenberg operators at site $i$. We rewrite the model in terms of bosonic magnon $a_i$ and fermionic holon $h_i$ operators by means of Holstein-Primakoff (HP) and slave-fermion transformations, respectively. For details expressions see Eqs. (9-10) in Appendix A or Ref. [8]. This leads to the following fermion-boson model:

$$\begin{aligned}
\mathcal{H} = &\ t \sum_{\langle i,j \rangle} h_i^\dagger h_j P_i \left( a_i + a_j^\dagger \right) P_j + \text{H.c.} \\
&+ \frac{J}{2} \sum_{\langle i,j \rangle} h_i h_i^\dagger \left[ P_i P_j a_i a_j + a_i^\dagger a_j^\dagger P_i P_j \right] h_j h_j^\dagger \\
&+ \frac{J}{2} \sum_{\langle i,j \rangle} h_i h_i^\dagger \left( a_i^\dagger a_i + a_j^\dagger a_j - 2\lambda a_i^\dagger a_i a_j^\dagger a_j - 1 \right) h_j h_j^\dagger,
\end{aligned} \tag{2}$$

where $P_i \equiv 1 - a_i^\dagger a_i$ [38]. The above model with $\lambda = 1$ follows from the *exact* mapping of the $t$–$J$ model. However, we also extend our discussion to the modified 1D $t$–$J$ model with $\lambda \neq 1$ so as to understand the effects of tuning the strength of the magnon-magnon interaction. We solve the above model numerically using Lanczos algorithm [39].

Naively one might have some doubts about using the magnon language to describe a 1D critical problem. Let us make two comments on this issue:

From the formal point of view there is nothing wrong with such approach—provided that the constraint on the number of bosons, always present in the slave-fermion transformation [8], is rigorously employed. (This is indeed done in all calculations below.) This statement can also be reformulated by stating that the magnons are here expressed in terms of hard-core bosons—in which case the constraint on number of bosons need not be employed.

On the other hand, employing the magnon language in 1D can give rise to new insights. First, it allows us for comparison of the 1D and 2D cases—for for the latter case it is the magnon language that is typically used to describe the low-energy excitations. In fact, this is the program that some of us adopted in the past to study the problem of a single hole in the Ising limit of the 1D $t$–$J$ model [40]: quite surprisingly in that case the linear spin wave theory breaks down and the magnon-magnon interaction are crucial to explain the destruction of the string potential and the ladder spectrum in 1D $t$–$J^z$ model [41]. We expect that at least some of these interesting results to carry on once the spin flips are included.

## 3   Results: switching on and off the magnon-magnon interaction

### 3.1   Ground state

We begin by studying the influence of the magnon-magnon interaction on the magnetic properties of the ground state of the fermion-boson model (2) with a single hole, cf. Fig. 1(a) vs. Fig. 1(b). To this end we choose the following three-point correlation function

$$\mathcal{C}(s,d) = (-1)^d 4L \langle S_0^z (1 - \tilde{n}_{s+d/2}) S_d^z \rangle. \tag{3}$$

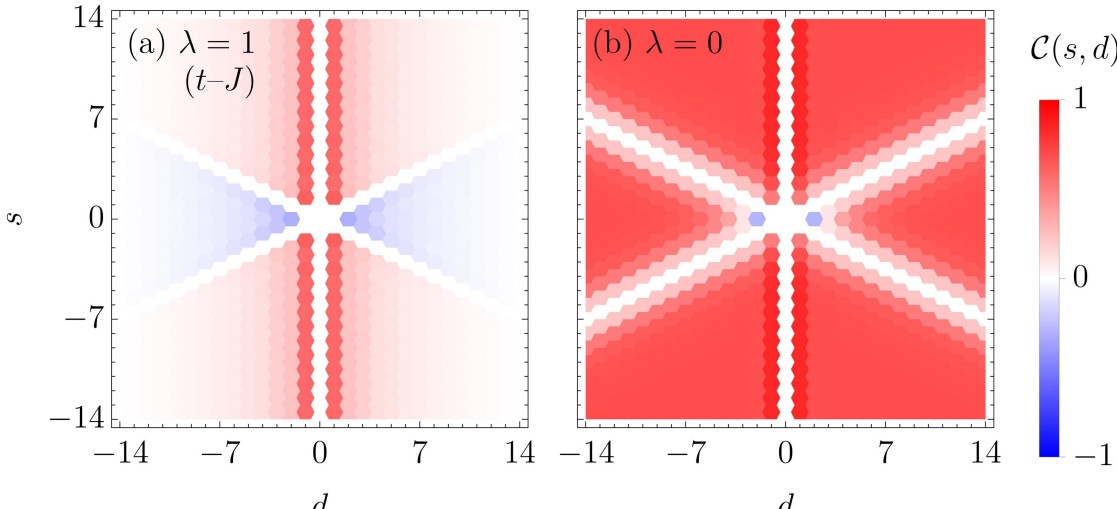

Figure 1: Magnetic properties of the fermion-boson model (2) ground state with a single hole as probed by the hole-spin correlation function $\mathcal{C}(s, d)$: (a) with magnon-magnon interaction 'correctly' included, i.e. with their value as in the 1D $t$–$J$ model [model (2) with $\lambda = 1$], (b) without the magnon-magnon interaction [model (2) with $\lambda = 0$]. Calculation performed on a 28 sites long periodic chain using exact diagonalization and for $J = 0.4t$, see text for further details.

Here, $d$ denotes the distance between the two spins, $s$ is the distance of the hole from the center of mass of the two spins and $L$ is the number of sites. As shown in Ref. [42] this 'hole-spins' correlator tracks the sign changes of the spin correlations due to the presence of the hole and hence can be used to verify whether spin-charge separation occurs in the system. Indeed, for the 1D $t$–$J$ model, i.e. once the parameter governing magnon-magnon attraction is tuned to the value of $\lambda = 1$ in the fermion-boson model (2), we fully recover the result of Ref. [42] and as shown in Fig. 1(a), the positive and negative correlation regimes are separated and extend to the largest accessible distance, reflecting the spin-charge separation nature. This contrasts with the hole-spins correlator calculated for the fermion-boson model (2) with $\lambda = 0$. Once the magnon-magnon interaction is switched off, cf. Fig. 1(b), the negative correlation is restricted to a very small regime with small $d$, indicating that the spinon and holon cannot be arbitrarily far apart. This sign structure of the hole-spins correlator is a signature of the spin polaron.

To irrevocably verify the stability of the spin polaron in a 1D antiferromagnet without magnon-magnon interaction, we perform a finite-size scaling analysis of the two crucial quantities defining the quasiparticle properties of the ground state: (i) the energy gap ($\Delta E$) between the ground and first excited states (at the same pseudomomentum $k = \pi/2$), and (ii) the quasiparticle spectral weight ($z$), i.e. the overlap between the ground state and the corresponding 'Bloch wave' single particle state.

To obtain the value of the energy gap $\Delta E$ in the thermodynamic limit we assume that $\Delta E$ scales linearly, up to a small logarithmic correction, as a function of the inverse system size $1/L$ [2] The finite size scaling analysis on the 1D $t$–$J$ model unambiguously shows that the energy gap quickly decreases with increasing system size and we obtain a vanishing $\Delta E$ in the thermodynamic limit within $10^{-2}t$ accuracy, cf. Fig. 2(a). This scaling behavior is consistent with the appearance of a low-energy continuum, which has been well demonstrated by exact diagonalization of the $t$–$J$ model. This result for the 1D $t$–$J$ model stands in stark contrast with the one

---

[2]This is due to: (i) the mapping of the problem of a single hole in the $t$–$J$ model onto a Heisenberg model with the shifted boundaries, cf. [41], and (ii) the energy gaps scaling in the latter model as $1/L$ with a small logarithmic correction, cf. [43].

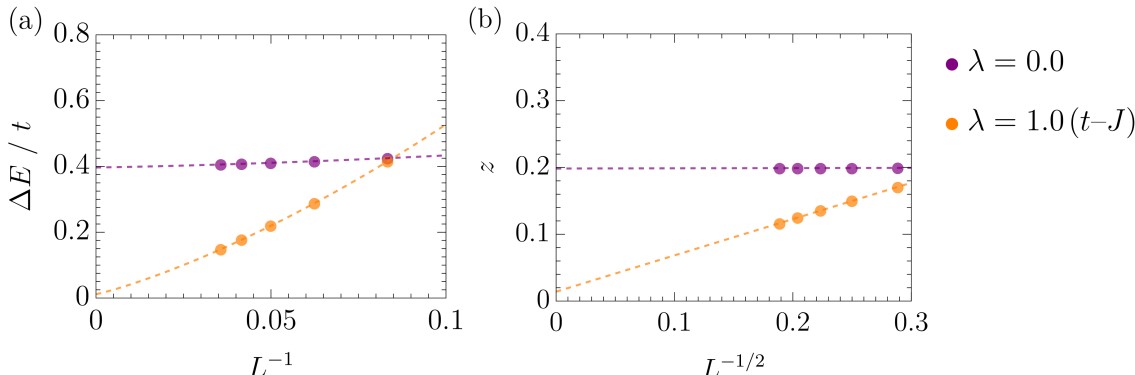

Figure 2: Dependence of the ground state quasiparticle properties in the fermion-boson model (2) with a single hole with system size $L$: (a) the energy difference $\Delta E$ between the ground state and the first excited state at the same pseudomentum $k = \pi/2$; (b) the quasiparticle spectral weight $z$, i.e. the overlap between the ground state and the 'Bloch wave' single particle state. Results with the magnon-magnon interaction correctly included in the 1D $t$–$J$ model [$\lambda = 1$ in (2)] are shown using orange symbols. Values of the magnon-magnon interaction for the modified $t$–$J$ model case $\lambda = 0$ (i.e. no magnon-magnon interaction) is shown using purple symbols. Calculation performed on chains of length $L$ and with $J = 0.4t$; see text for details on the finite-size-scaling functions fitted to the data.

obtained for the modified $t$–$J$ model with switched off magnon-magnon interaction ($\lambda = 0$); in that case the energy gap $\Delta E$ scales to a finite value, cf. Fig. 2(a), consistent with the quasiparticle picture.

We also calculated the quasiparticle spectral weight $z$ in the thermodynamic limit, cf. Fig. 2(b), by assuming that it scales as $1/\sqrt{L}$ with system size $L$, based on the exact result known for the 1D $t$–$J$ model [3]. We again obtain strongly contrasting behaviors: in the 1D $t$–$J$ case [i.e. $\lambda = 1$ in (2)], $z$ vanishes asymptotically within $10^{-2}$ numerical accuracy, further confirming the absence of a quasiparticle. On the other hand, for $\lambda = 0$ $z$ converges to a finite value—for instance $z \approx 0.2$ for $J = 0.4t$.

## 3.2   Excited states

The impact of magnon-magnon interaction should not only be restricted to the low-energy quasiparticle but also extend to the distribution of the high-energy excited states. Therefore, we calculate the single particle spectral function of the fermion-boson model (as measured by ARPES) both at the critical value $\lambda = 1$ and for $\lambda = 0$ (in the next section we will also vary the strength of the magnon-magnon interaction beyond these two specific values):

$$A(k, \omega) = -\frac{1}{\pi} \operatorname{Im} G(k, \omega + i0^+), \tag{4}$$

$$G(k, \omega) = \langle \psi_{\mathrm{GS}} | \tilde{c}_k^\dagger \frac{1}{\omega - \mathcal{H} + E_{\mathrm{GS}}} \tilde{c}_k | \psi_{\mathrm{GS}} \rangle, \tag{5}$$

where $|\psi_{\mathrm{GS}}\rangle$ and $E_{\mathrm{GS}}$ stand for ground state wave function of the antiferromagnetic Heisenberg model and its ground state energy respectively, and $\tilde{c}_k = (\tilde{c}_{k\uparrow} + \tilde{c}_{k\downarrow})/\sqrt{2}$. Note that replacing $\tilde{c}_k \to \tilde{c}_{k\sigma}$ does not affect the result. Rewriting $G(k, \omega)$ in terms of the fermion-boson model

---

[3]As per exact result obtained for the 1D $t$–$J$ model with $J = 2t$ and for 'ground state' momentum $p = \pi/2$, cf. [44].

operators, we obtain,

$$G(k, \omega) = \frac{1}{2N} \sum_{i,j} \langle \psi_{\text{GS}}^{\text{fb}} | (1 + a_j^\dagger) P_j h_j \frac{e^{-ik(r_i - r_j)}}{\omega - \mathcal{H} + E_{\text{GS}}} h_i^\dagger P_i (1 + a_i) | \psi_{\text{GS}}^{\text{fb}} \rangle . \qquad (6)$$

Here $|\psi_{\text{GS}}^{\text{fb}}\rangle$ and $|\psi_{\text{GS}}\rangle$ are related by a rotation of one sublattice and slave-fermion transformation.

The results are shown in Fig. 3(a-b). The spectrum for $\lambda = 1$ is identical to the well-known spectral function of the $t$–$J$ model at half-filling [18,45], cf. Fig. 3(a). The incoherent spectrum is usually understood in terms of a convolution of the spinon and holon dispersion relations [shown by the dashed lines in Fig. 3(a)].

The spectrum in the absence of the magnon-magnon interaction, i.e. at $\lambda = 0$, is shown in Fig. 3(b). This spectrum contains a dispersive low-energy feature which is visibly split from the rest of the spectrum at momenta $k > \pi/2$ and which, at $k = \pi/2$, corresponds to the spin polaron quasiparticle characterized in Figs. 2. Crucially, the whole spectrum exhibits typical features of the spin polaron physics. To verify that this is the case, we qualitatively reproduced the result of Fig. 3(b) using a self-consistent Born approximation (not shown) to the spectral function of the modified 1D $t$–$J$ model at $\lambda = 0$ [4], i.e. using an 'archetypical' spin polaronic calculation.

Interestingly, *apart* from the dispersive low-energy quasiparticle feature particularly pronounced for $k > \pi/2$, the two spectra seem to be qualitatively similar. This stunning result originates from the fact that: (i) excited states with a predominantly moderate number of sparsely distributed magnon pairs have an important contribution to the excited states of model (2) at any $\lambda$, (ii) for such states the magnon-magnon interaction do not matter, hence they contribute in a similar manner to the spectral function for any $\lambda$, in particular $\lambda = 1$ and $\lambda = 0$.

These results enable us to give an alternative, albeit approximate, understanding of the dominant features appearing at $\omega \propto t |\cos k|$ in the spectrum at $\lambda = 1$. These dispersions are well accounted for in the spin-charge separation picture as the 'free' holons, cf. [47,48] and dashed lines of Fig. 3(a-b). Here, based on the similarity between $\lambda = 1$ and $\lambda = 0$ spectra, we can approximately interpret the two dominant spectral features as being due to a holon propagating in a polaronic way by exciting a single magnon (Born approximation) at a vertex $t |\cos k|$.

## 4 Results: tuning the value of the magnon-magnon interaction

A striking feature of the fermion-boson model (2) is that, at the qualitative level, the spin polaron solution to the single hole problem dictated by (2) exists not only when the magnon-magnon attraction is switched off but also for all values of the magnon-magnon attraction *except* for the 'critical' $\lambda = 1$, which preserves the SU(2) symmetry of the spin interactions [the SU(2) symmetry is broken in the model once $\lambda \neq 1$ in (2), see Appendix B for details]. This result is visible when looking at the observables used above for values of magnon-magnon interaction $\lambda$ other than 0 or 1:

First, we present below the results for the three-point correlation function $C(s, d)$ [defined in Eq. (3)] for the intermediate value of magnon-magnon interaction $\lambda = 0.5$ as well as $\lambda = 0.9$ and $\lambda = 1.1$, which are 'close' to ideal $t$–$J$ model case (i.e. $\lambda = 1.0$)—see Fig. 4. Even for $\lambda$ close to 1, it is very clear that the cloud of magnetic excitations (flipped spins) can be observed in a small region around a hole. Such a picture is a signature of a spin polaron and shows the breakdown of the spin-charge separation once $\lambda \neq 1$. This result can be further confirmed by checking how quasiparticle properties of the fermion-boson model (2) ground state vary with the magnon-magnon interaction, see Fig. 5. We observe that the energy gap $\Delta E$ as well as the

---

[4] We use a small but finite Ising anisotropy, in order to overcome the divergence of the linear spin wave theory and assume unbroken SU(2) symmetry [46].

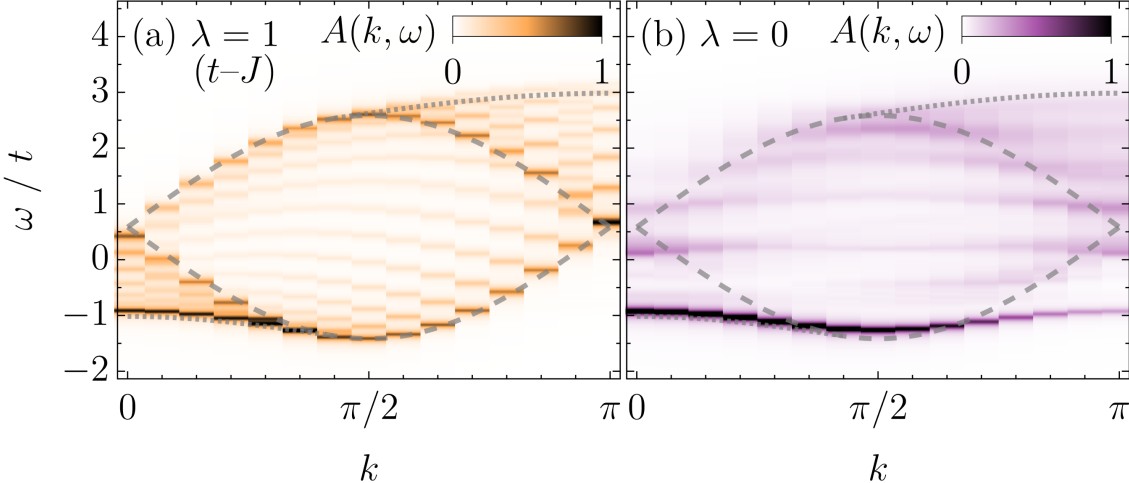

Figure 3: Properties of the excited state of the fermion-boson model (2) with a single hole as probed by the spectral function $A(k,\omega)$: (a) with the magnon-magnon interaction correctly included [1D $t$–$J$ model, $\lambda = 1$ in (2)]; (b) without the magnon-magnon interaction [$\lambda = 0$ in (2)]. The dashed (dotted) lines in (a-b) show the holon (spinon) dispersion relations respectively, as obtained from the spin-charge separation Ansatz [47,48]. The highest intensity peak at lowest energy in (b) is the spin polaron quasiparticle peak. Calculation performed on a 28 sites long periodic chain using exact diagonalization and with $J = 0.4t$.

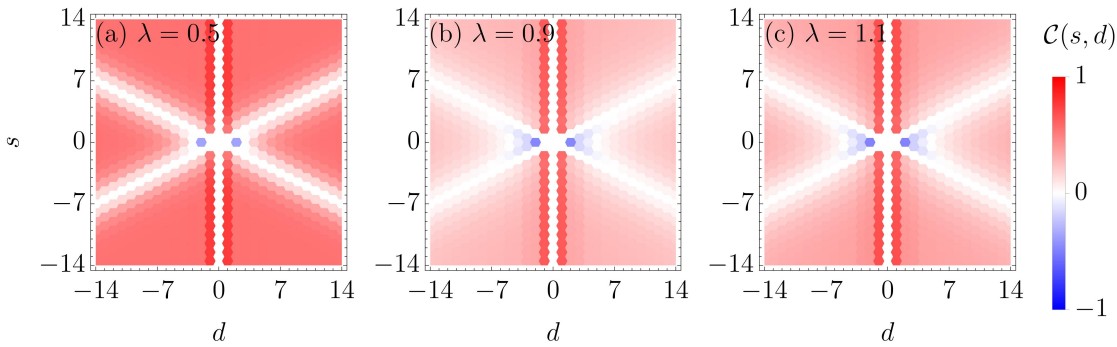

Figure 4: Magnetic properties of the fermion-boson model (2) ground state with a single hole as probed by the hole-spin correlation function $\mathcal{C}(s,d)$: (a) with an 'intermediate' value of magnon-magnon interaction $\lambda = 0.5$, (b-c) with the magnon-magnon interaction $\lambda = 0.9$ and $\lambda = 1.1$ 'close' to the ideal $t$–$J$ model case ($\lambda = 1.0$). Calculation performed on the $L$ sites long periodic chain using exact diagonalization and for $J = 0.4t$.

quasiparticle spectral weight remains finite even once $\lambda$ is close to one—but *not* exactly equal to one. Altogether, this shows that the spin polaron quasiparticle solution is stable once the magnon-magnon interaction are tuned away from their value given by the 1D $t$–$J$ model.

Second, we investigate how the properties of the excited states of the fermion-boson model (2) change once the magnon-magnon interaction is tuned, see Fig. 6. Just as for the ground state, also the spectral function $A(k,\omega)$ is qualitatively the same as soon as the value of the magnon-magnon interaction is tuned away from its value in the 1D $t$–$J$ model.

In order to obtain an even more intuitive understanding of the crucial role played by the specific value of the magnon-magnon interaction, as well as to connect with the results for the $t$–$J^z$ model of [40], we introduce one more observable: The probability $c_n$ of finding a state with $n$ magnons forming a chain attached to one side of the single hole in the ground state of (2), see Fig. 7(b)

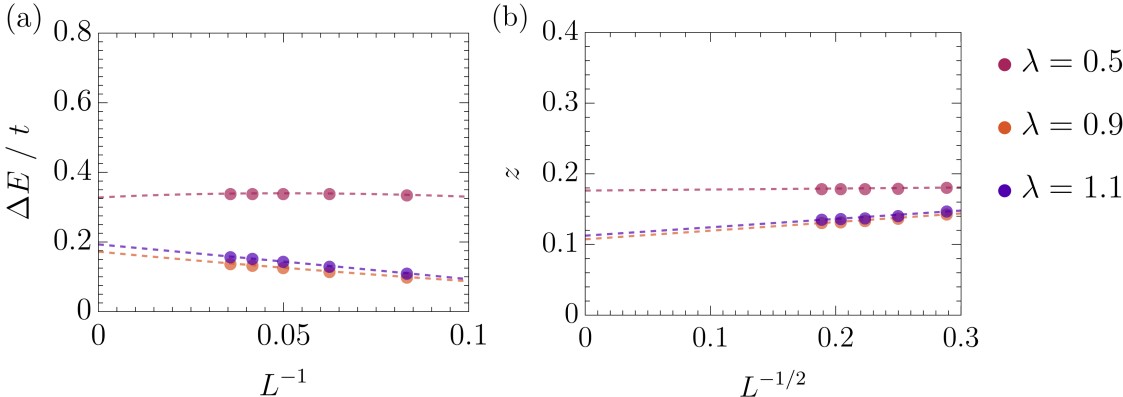

Figure 5: Dependence of the ground state quasiparticle properties in the fermion-boson model (2) with a single hole with system size $L$: (a) the energy difference $\Delta E$ between the ground state and the first excited state at the same pseudomentum $k = \pi/2$; (b) the quasiparticle spectral weight $z$, i.e. the overlap between the ground state and the 'Bloch wave' single particle state. Results obtained for the modified $t$–$J$ model case in (2)] with the magnon-magnon interaction $\lambda = 0.5$, $\lambda = 0.9$ and $\lambda = 1.1$. Calculation performed on chains of length $L$ and with $J = 0.4t$; see text for details on the finite-size-scaling functions fitted to the data.

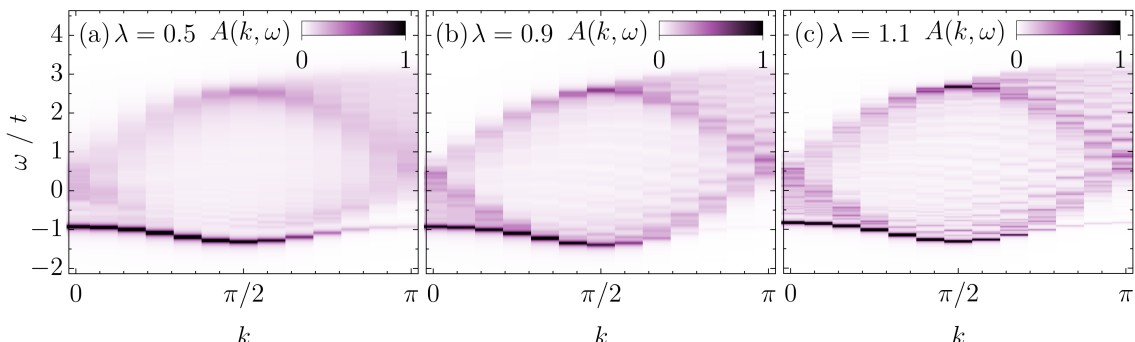

Figure 6:  Properties of the excited state of the fermion-boson model (2) with a single hole as probed by the spectral function $A(k, \omega)$ with different values of the magnon-magnon interaction: (a) $\lambda = 0.5$, (b) $\lambda = 0.9$ , (c) $\lambda = 1.1$ in (2). The highest intensity peak at lowest energy in (a-c) is the spin polaron quasiparticle peak. Calculation performed on a 24 sites long periodic chain using exact diagonalization and with $J = 0.4t$.

for a pictorial view of this observable. The probabilities $c_n$ for various values of the magnon-magnon interaction are shown in Fig. 7(a). The first result here is that only at the critical value of the magnon-magnon interaction $\lambda = 1$ the $c_n$'s are almost the same for all $n$, consistent with spin-charge separation, cf. Fig. 7(a). This is because, at $\lambda = 1$ only, the cost of creating an extra magnon next to an existing magnon is precisely cancelled by their attraction. Hence, none of the magnons created by the mobile hole costs any energy apart from the first one, as long as they form a string. This, together with the magnon pair creation and annihilation terms [terms $\propto a_i a_j + h.c.$ in Eq. (2)], allows for almost constant $c_n$'s in the bulk of the chain.

    Once $\lambda \neq 1$ the probability $c_n$ is *never* a constant function of $n$ and spin-charge separation cannot take place [cf. Fig. 7(a), *inter alia* note the distinct behavior for $\lambda = 0.99$ and $\lambda = 1$]. This is due to the fact that for $\lambda \neq 1$ there can never be an exact 'cancellation' between the on-site magnon energy and the interaction one. In particular, for the physically interesting case of $0 \leq \lambda < 1$, that interpolates between the exact expression for the 1D $t$–$J$ model and the linear spin-

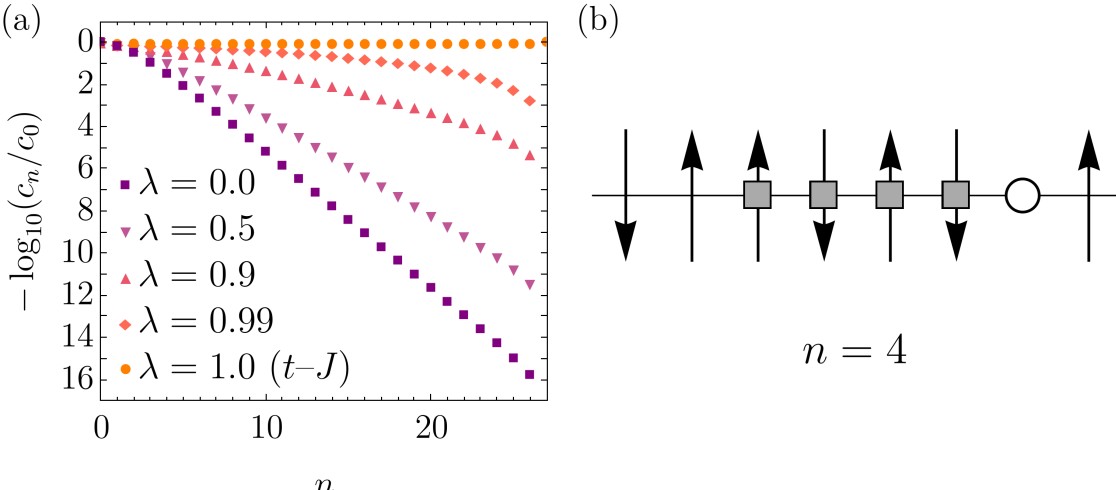

Figure 7: Properties of the fermion-boson model (2) with a single hole and with different strength of the magnon-magnon interaction $\lambda$. Panel (a) shows probabilities $c_n$ of finding a configuration with $n$ consecutive magnons attached to one side of the hole in the ground state of the respective model; panel (b) shows a pictorial view of a configuration with $n = 4$ magnons attached to the left side of the hole; All data obtained using exact diagonalization on a 28 sites periodic chain using $J = 0.4t$.

wave approximation, $c_n$ decreases superexponentially with increasing the number of magnons $n$, cf. Fig. 7(a). This is due to the mobile hole exciting magnons whose energy cost grows linearly with their number. Hence, the total energy is optimised through a subtle competition between the hole polaronic energy and the magnon energy leading to the superexponentially suppressed probability of finding a configuration with an increasing number of magnons. This signals the onset of the string potential and the spin polaron picture, as discussed in detail in the context of the 2D $t$–$J^z$ model in Ref. [40].

## 5 Discusssion: relevance for real materials

The existence of just one critical value of the magnon-magnon interaction [$\lambda = 1$ in (2)] stabilising the spin-charge separation solution, and at the same time onset of the spin polaron solution for all other values of the magnon-magnon interaction, is a striking feature of the fermion-boson model (2). Interestingly, this rather abstract result, has an important consequence for real materials.

Due to the nature of atomic wavefunctions and crystal structures, the best-known '1D' antiferromagnetic materials (cf. $Sr_2CuO_3$, $SrCuO_2$, or $KCuF_3$) are solely *quasi*-1D [26–28]. A precise model of these materials should include a small but finite staggered magnetic field $J_\perp$ [see Appendix A for details], which originates in the magnetic coupling between the spins on neighboring chains [49–51]. Importantly, the single-hole dynamics in a 1D $t$–$J$ model with staggered field is qualitatively the same (and quantitatively very similar, as discussed in the Appendix A) as that in a 1D $t$–$J$ model with modified magnon-magnon interaction. Indeed, the strength of the staggered field can be mapped to that of the magnon-magnon interaction, see Appendix A. Altogether this means that the presence of the staggered field disrupts the above-discussed fine balance between the on-site magnon energy and the magnon-magnon attraction seen in the 1D $t$–$J$ model. Therefore, the mobile hole in the *quasi*-1D cuprates experiences the string potential and forms the spin polaron, cf. Fig 8(a). This indicates that spin-charge separation is, strictly speaking, never realised in real materials.

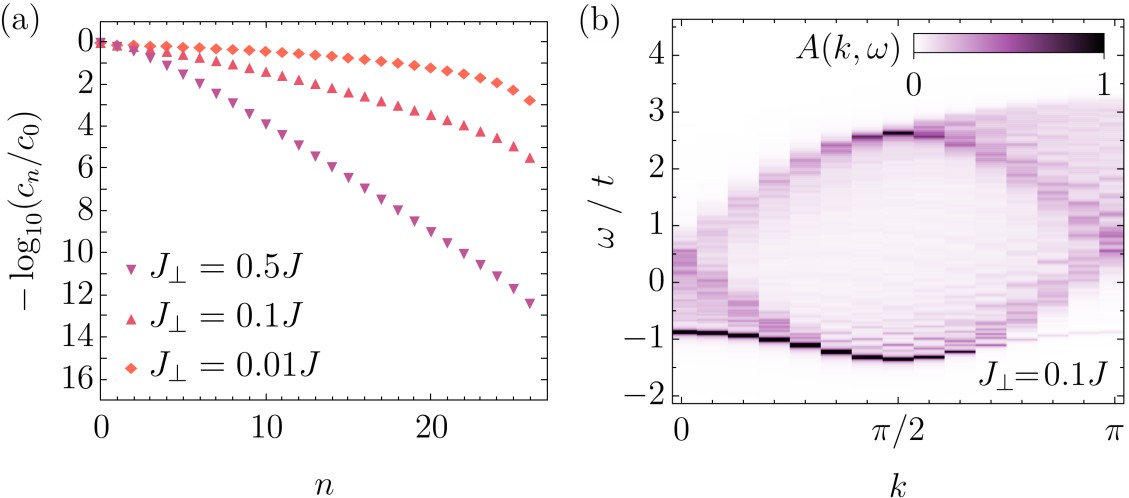

Figure 8: Properties of 1D $t$–$J$ model (1) with a single hole *and* with added staggered magnetic field arising due to the coupling [$J_\perp/J$ in Eq. (7) in Appendix A] to neighboring chains in a quasi-1D geometry, cf. text and Appendix A. Panel (a) shows probabilities $c_n$ of finding a configuration with $n$ consecutive magnons attached to one side of the hole in the ground state of the respective model; panel (b) shows the spectral function $A(k,\omega)$ calculated for the 1D $t$–$J$ model with added staggered magnetic field $J_\perp = 0.1J$. All data obtained using exact diagonalization on a 28 sites periodic chain using $J = 0.4t$.

One may wonder how to reconcile the above finding with the fact that ARPES experiments on *quasi*-1D cuprates have reported spin-charge separation [18–20, 22, 23] based on the experimentally measured spectrum being similar to the one obtained for the 1D $t$–$J$ model [$\lambda = 1$ in Eq. (2)], cf. Fig. 3(a) [18–20, 22, 23]. The salient fact is that the spectrum obtained when a small staggered field $0 < J_\perp \lesssim 0.1J$ acts on the 1D $t$–$J$ model, cf. Fig 8(b), is *almost indistinguishable* from the one of Fig. 3(a), especially when we consider the ARPES resolution. In fact, for the available finite size calculations with the numerical broadening $\delta = 0.05t$, the only visible difference between the two spectra lies in an extremely faint quasiparticle feature present for $k > \pi/2$. Since the latter feature cannot be discerned with the current ARPES resolution, especially at high temperature and with a typically weaker signal for $k > \pi/2$ in ARPES, we conclude that ARPES measurements on *quasi*-1D cuprates are more correctly interpreted using the spin polaron picture, with its dominant cosine-like features interpreted as the holon exciting a magnon at a vertex $\propto t|\cos k|$ (see above).

## 6  Summary

In this work we discussed the extent to which the concept of the spin polaron, well-known from the studies of a single hole in 2D antiferromagnets [52], can be applied to the single hole problem in the 1D antiferromagnets. We find that *only* in the 1D SU(2) symmetric limit the spin polaron is unstable to spin-charge separation due to the critical value of the magnon-magnon interaction. In contrast, the spin polaron quasiparticle is stable in the real *quasi*-1D antiferromagnets such as $SrCuO_2$, $Sr_2CuO_3$ or $KCuF_3$.

The obtained results are completely in line with the concept which stems from the boson-fermion toy-model calculations of Ref. [53]: these suggest that a fermionic quasiparticle *may* collapse once an electron is coupled to bosons whose energy reaches zero for *some* of the sites (while the converse is not true). On the other hand, the surprising robustness of the spin polaron leaves us with an *open question* whether this simple picture can be used to study also the higher-

dimensional highly-doped antiferromagnets *beyond* the collapse of the long-range order.

# Acknowledgements

We thank Federico Becca, Sasha Chernyshev, Mario Cuoco, Alberto Nocera, and Steve Johnston for stimulating discussions.

For the purpose of Open Access, the authors have applied a CC-BY public copyright licence to any Author Accepted Manuscript (AAM) version arising from this submission.

**Funding information**   We kindly acknowledge support by the (Polish) National Science Centre (NCN, Poland) under Projects No. 2016/22/E/ST3/00560, 2016/23/B/ST3/00839, 2021/40/C/ST3/00177 as well as the Excellence Initiative of the University of Warsaw ('New Ideas' programme) IDUB program 501-D111-20-2004310 "Physics of the superconducting copper oxides: 'ordinary' quasiparticles or exotic partons?". Y.W. acknowledge support from the National Science Foundation (NSF) award DMR-2132338. M.B. acknowledges support from the Stewart Blusson Quantum Matter Institute and from NSERC. K.W. thanks the Stewart Blusson Quantum Matter Institute for the kind hospitality. This research was carried out with the support of the Interdisciplinary Center for Mathematical and Computational Modeling at the University of Warsaw (ICM UW) under grant no G73-29.

## A The $t$–$J$ model for *quasi*-1D cuprates: tunable staggered magnetic field vs. tunable magnon-magnon interaction

In order to construct the $t$–$J$ model for *quasi*-1D cuprates, we make the following assumptions:

*First*, to get qualitative insight into the hole motion, we note that hopping between the chains can be neglected [54] and that the longer-range hopping is very small for *quasi*-1D cuprates [55]. Besides, the recently postulated strong coupling to phonons in 1D cuprates [56], not included here, would only further disrupt the (mentioned below and in the main text of the paper) fine balance between the magnon-magnon interaction and the magnon on-site energy.

*Second*, the remaining Heisenberg exchange interaction between the chains can be represented as the staggered magnetic field (which, due can be obtained from the spin exchange between the chains [49], hence is called $J_\perp$ below and in the main text of the paper):

$$H_{J_\perp} = \frac{J_\perp}{2} \sum_{\langle i,j \rangle} \left[ (-1)^i S_i^z + (-1)^j S_j^z \right]. \tag{7}$$

The above term follows by assuming the onset of the long-range magnetic order at low temperatures, the magnetic interactions between the chains can be treated on a mean-field level—which, irrespectively of the sign of the interchain coupling, yields a staggered magnetic field acting on the antiferromagnetic chain in which the hole moves [49–51]. We note that, also at higher temperatures, i.e. when there is no long-range order and the staggered field cannot be used to simulate the coupling between the chains, the (mentioned in the main text of the paper) fine balance between the magnon-magnon interaction and their onsite energies will *also* be disrupted due to the change of magnon on-site energies by the exchange interaction between the chains. Thus, all of the presented results, obtained below will qualitatively model the *quasi*-1D cuprates also at temperatures higher than the Neel temperature. (We do not present such calculations, since they require exact diagonalisation of a full 2D problem, which heavily suffers from finite size effects and is beyond the scope of this work.) Following [49] one can estimate the value of $J_\perp$ in various *quasi*-1D cuprates: e.g. for KCuF$_3$ we obtain $J_\perp \approx 0.06J$ [hence the assumed in Fig. 4(d) of the main text value $J_\perp = 0.1J$, being the upper bound of that estimate].

Now let us investigate how the additional staggered field looks like in the polaronic description already used in the main text. In order to do this we firstly show in detail the polaronic descritpion of the 1D $t$–$J$ model [i.e. how to go from Eq. (1) to Eq. (2) of the main text]. To this end, we start with a rotation of spins in one of the system's sublattices. This results in

$$H_{\text{rot}} = -t \sum_{\langle i,j \rangle, \sigma} \left( \tilde{c}_{i\sigma}^\dagger \tilde{c}_{j\bar{\sigma}} + H.c \right) + J \sum_{\langle i,j \rangle} \left[ \frac{1}{2} \left( S_i^+ S_j^+ + S_i^- S_j^- \right) - S_i^z S_j^z - \frac{1}{4} \tilde{n}_i \tilde{n}_j \right]. \tag{8}$$

This allows for the introduction of holes and magnons according to the following transformations

$$\begin{aligned} \tilde{c}_{i\uparrow}^\dagger &= P_i h_i, & \tilde{c}_{i\uparrow} &= h_i^\dagger P_i, \\ \tilde{c}_{i\downarrow}^\dagger &= a_i^\dagger P_i h_i, & \tilde{c}_{i\downarrow} &= h_i^\dagger P_i a_i, \end{aligned} \tag{9}$$

$$\begin{aligned} S_i^+ &= h_i h_i^\dagger P_i a_i, & S_i^z &= \left( \frac{1}{2} - a_i^\dagger a_i \right) h_i h_i^\dagger, \\ S_i^- &= a_i^\dagger P_i h_i h_i^\dagger, & \tilde{n}_i &= 1 - h_i^\dagger h_i = h_i h_i^\dagger, \end{aligned} \tag{10}$$

where $a_i^\dagger$ are bosonic creation operation at site $i$ denoting magnons and $h_i^\dagger$ are fermionic creation operators at site $i$ denoting holons. Operator $P_i$ projects onto a subspace with 0 magnons at site $i$. Here magnons can be understood as deviations from the state that has all the spins pointing

up after the applied sublattice rotation. In the end, the 1D $t-J$ model (up to a shift by constant energy) reads:

$$\mathcal{H} = \mathcal{H}_t + \mathcal{H}_J, \tag{11}$$

where,

$$\mathcal{H}_t = t \sum_{\langle i,j \rangle} \left\{ h_i^\dagger h_j P_i \left[ a_i + a_j^\dagger \right] P_j + h_j^\dagger h_i P_j \left[ a_j + a_i^\dagger \right] P_i \right\}, \tag{12}$$

$$\begin{aligned}
\mathcal{H}_J &= \frac{J}{2} \sum_{\langle i,j \rangle} h_i h_i^\dagger \left[ P_i P_j a_i a_j + a_i^\dagger a_j^\dagger P_i P_j \right] h_j h_j^\dagger \\
&+ \frac{J}{2} \sum_{\langle i,j \rangle} h_i h_i^\dagger \left( a_i^\dagger a_i + a_j^\dagger a_j - 2 a_i^\dagger a_i a_j^\dagger a_j - 1 \right) h_j h_j^\dagger.
\end{aligned} \tag{13}$$

Now let us investigate the staggered magnetic field term given by Eq. (7) above. Performing the same set of transformations we obtain (up to a constant energy shift),

$$\mathcal{H}_{J_\perp} = \frac{J_\perp}{2} \sum_{\langle i,j \rangle} \left( a_i^\dagger a_i h_i h_i^\dagger + a_j^\dagger a_j h_j h_j^\dagger \right) \approx \frac{J_\perp}{2} \sum_{\langle i,j \rangle} h_i h_i^\dagger \left( a_i^\dagger a_i + a_j^\dagger a_j \right) h_j h_j^\dagger. \tag{14}$$

The omitted terms on the right hand side of the approximation modify the magnetic field only around the hole and they are $\propto J_\perp \left( a_i^\dagger a_i h_j h_j^\dagger + a_j^\dagger a_j h_i h_i^\dagger \right)$. In the end, we obtain for the spin part of the Hamiltonian [$\mathcal{H}_t$ is not affected, i.e. given by Eq. (12) above]

$$\begin{aligned}
\mathcal{H}_{J+J_\perp} &\equiv \mathcal{H}_J + \mathcal{H}_{J_\perp} \\
&\approx \frac{J}{2} \sum_{\langle i,j \rangle} h_i h_i^\dagger \left[ P_i P_j a_i a_j + a_i^\dagger a_j^\dagger P_i P_j \right] h_j h_j^\dagger \\
&+ \frac{J}{2} \sum_{\langle i,j \rangle} h_i h_i^\dagger \left[ \left( 1 + \frac{J_\perp}{J} \right) \left( a_i^\dagger a_i + a_j^\dagger a_j \right) - 2 a_i^\dagger a_i a_j^\dagger a_j - 1 \right] h_j h_j^\dagger.
\end{aligned} \tag{15}$$

Let us introduce the XXZ anisotropy

$$\Delta = \frac{J_\perp}{J} \tag{16}$$

and the rescaled magnon-magnon interaction parameter

$$\lambda = \frac{1}{1 + \Delta}. \tag{17}$$

Then, in the single hole limit, we can write

$$\begin{aligned}
\mathcal{H}_{J+J_\perp} &\approx \frac{J}{2} \sum_{\langle i,j \rangle} h_i h_i^\dagger \left[ P_i P_j a_i a_j + a_i^\dagger a_j^\dagger P_i P_j \right] h_j h_j^\dagger \\
&+ (1 + \Delta) \frac{J}{2} \sum_{\langle i,j \rangle} h_i h_i^\dagger \left( a_i^\dagger a_i + a_j^\dagger a_j - 2 \lambda a_i^\dagger a_i a_j^\dagger a_j \right) h_j h_j^\dagger.
\end{aligned} \tag{18}$$

Thus, once $J_\perp \neq 0$ the final model is the $t-J$ model with the XXZ anisotropy $\Delta$ *and* rescaled magnon-magnon interaction $\lambda$. In TABLE 1. we present the values of $\lambda$, $\Delta$ calculated for the corresponding values of $J_\perp$ used in calculations for Fig. 4(b) and 4(d) in the main text.

| $J_\perp/J$ | $\Delta$ | $\lambda$ |
|:---:|:---:|:---:|
| 0.01 | 0.01 | $\frac{100}{101}$ |
| 0.1 | 0.1 | $\frac{10}{11}$ |
| 0.5 | 0.5 | $\frac{2}{3}$ |

Table 1: Table presenting the relation between the value of the staggered field $J_\perp$ in the *quasi*-1D $t$–$J$ model and the $t$–$J$ model with rescaled magnon-magnon interaction $\lambda$ and the XXZ anisotropy $\Delta$.

# B SU(2) symmetry breaking in the $t$–$J$ model with tunable magnon-magnon interaction

We start by re-expressing the magnon-magnon interaction term in the 'standard' (i.e. spin) language,

$$a_i^\dagger a_i a_j^\dagger a_j = -S_i^z S_j^z + \frac{1}{4}\tilde{n}_i\tilde{n}_j - \frac{1}{2}\left(\xi_i^\mathcal{A} S_i^z + \xi_j^\mathcal{A} S_j^z\right)\tilde{n}_i\tilde{n}_j, \tag{19}$$

where $\xi_i^\mathcal{A}$ equals $-1$ for $i \in \mathcal{A}$ and $1$ otherwise, with $\mathcal{A}, \mathcal{B}$ denoting the two sublattices of the bipartite lattice. Thus, Hamiltonian (2) of the main text (i.e. the $t$–$J$ model with tuneable magnon-magnon interaction) reads,

$$H = -t\sum_{\langle i,j\rangle}\left(\tilde{c}_{i\sigma}^\dagger \tilde{c}_{j\sigma} + \text{H.c.}\right)$$
$$+ J\sum_{\langle i,j\rangle}\left\{S_i S_j - \frac{1}{4}\tilde{n}_i\tilde{n}_j + (\lambda-1)\left[S_i^z S_j^z - \frac{1}{4}\tilde{n}_i\tilde{n}_j + \frac{1}{2}\left(\xi_i^\mathcal{A} S_i^z + \xi_j^\mathcal{A} S_j^z\right)\tilde{n}_i\tilde{n}_j\right]\right\}. \tag{20}$$

In the above Hamiltonian (20), the term

$$\frac{1}{2}\left(\xi_i^\mathcal{A} S_i^z + \xi_j^\mathcal{A} S_j^z\right)\tilde{n}_i\tilde{n}_j \tag{21}$$

can be understood as a staggered field acting on all spins although it is halved for the neighbors of the hole. This term contributes to the Hamiltonian once $\lambda \neq 1$ and explicitly breaks the SU(2) symmetry.

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
