# Peer review of "The fate of the spin polaron in the 1D antiferromagnets"

_SciPost Physics_

## Round 4 · Referee Report · Anonymous (Referee 1) · 2023-10-20

Report
In terms of comparison with angle-resolved photoemission (ARPES) data for 1D Mott insulators, the authors emphasized that real materials do not obey the SU(2) t-J model but there are possible interactions that break the symmetry. Therefore, the concept of spin-charge separation is incomplete. This is a correct statement as long as one considers temperatures well below the energy scale of symmetry-breaking interactions.
Concerning the symmetry breaking, Figure 3 demonstrated the change of ARPES spectrum in terms of $\lambda$. In the main text, it was mentioned that "the two spectra seem to be qualitatively similar". However, the distribution of spectral weight seems to be different, for example, the number of stripes running along $k$ direction is deferent: for $\lambda=1$, the number is determined by the size of system, while for $\lambda=0$, the number is roughly determined by the strength of spin-polaron formation. Therefore, it is recommended to make clear which structures are similar each other in the spectra.
We thank the Referee for the very kind and constructive report. Below we address the following concern raised by the Referee:
"Concerning the symmetry breaking, Figure 3 demonstrated the change of ARPES spectrum in terms of λ. In the main text, it was mentioned that "the two spectra seem to be qualitatively similar". However, the distribution of spectral weight seems to be different, for example, the number of stripes running along k direction is deferent: for λ=1, the number is determined by the size of system, while for λ=0, the number is roughly determined by the strength of spin-polaron formation. Therefore, it is recommended to make clear which structures are similar each other in the spectra."
Before answering it in detail, just to avoid confusion, we would like to stress that the discussion below concerns only the excited states of the spectrum shown in Fig. 3: since, as also discussed in the paper, the low-energy part of the two spectra differ by "the dispersive low-energy quasiparticle feature particularly pronounced for k > π/2".
(1) The main similarity between the two spectra follows from the fact that both for λ=1 and λ=0 in Fig. 3:
— (most importantly) almost all the spectral weight is tightly enclosed by the dashed and dotted lines (indicating the dispersion of the free holon and the edges of the spinon-holon continuum),
— dashed lines track quite well the position of the enhanced spectral weight in the (k,\omega) plane (this is for all lines except for the lower-left dashed holon line),
(2) On the other hand, the main difference between the λ=1 and λ=0 in Fig. 3 is related to the distinct nature of "stripes" in the spectrum (as very nicely pointed out by the Referee):
— for λ=1 the "stripes" indeed come from the finite size effect and disappear in the thermodynamic limit, merging into a continuum of states.
— for λ=0 the "stripes" are indeed determined by the strength of the string potential, as the "stripes" follow from the ladder spectrum.
(3) Nevertheless, we note that just as it is the case for the spin polaron spectrum of the 2D t-J model [without magnon-magnon interaction, i.e. as calculated by SCBA, cf. Fig. 4(b) of Phys. Rev. B 92, 075119 (2015)], the "string-potential" stripes should largely be washed for λ=0 in Fig. 3. (Note that the 1D t-J model at λ=0 contains the same qualitative physics as the 2D t-J treated with SCBA, cf. the discussion in the paper.) In other words, in the thermodynamic limit in both cases the main feature will be a continuum, whose intensity in the λ=0 case will be modulated by the stripes. Considering that experimentally measured spectra are broadened because of a variety of factors, we think it is unlikely that this fine structure could be resolved experimentally. This is why we claim that the higher-energy part of the spectrum (the continuum) is likely to look quite similar.
We expand the discussion in the revised version of the manuscript along these lines (see summary of changes below).
We note that it has taken us some time to finalise the report, since, stimulated by the reports of the two other referees, we have decided to perform several additional calculations and present an even more detailed understanding of the spin polaron collapse. In particular, we now show how the collapse of the spin polaron in the 1D t-J model is related not only to the special value of the magnon-magnon interactions (which still holds) but also to the hard-core nature of the magnons. The latter allows for the hole to have a finite coupling to gapless magnons even at zero energy and momentum transfer. We believe that with this additional deep insight the paper is now fully self-contained and presents a significant advancement in our understanding of the single hole problem, that is worth publishing in SciPost Physics.

Author: Krzysztof Wohlfeld on 2024-06-16 [id 4571]
(in reply to Report 3 on 2023-11-06)We thank the Referee for the very kind and constructive report. Below we address the main concern raised by the Referee:
"This paper proposes a new way of attacking a - somehow - old problem, the instability of the quasiparticle weight in the 1D t-J model. The work is based on an exact boson-fermion rewriting of the 1D t-J model proposed by Peter Horsch et al. in the 90's. This provides new interesting perspectives but, I believe, no new findings not reported before."
Let us begin by clarifying a misunderstanding: Whereas indeed the works by Peter Horsch et al. in the 90s were fundamental in building up the "spin polaron theory", all of these works have been implicitly based on the linear spin wave theory and have always been applied to ground states with magnetic long range order. In this work, our aim is to use the same magnon-holon basis as Peter Horsch et al. but to solve a 1D problem without long-range order. Such an approach, which certainly requires goind beyond the linear spin wave theory, has never been done before.
The presented results are, in our opinion, completely novel. We explain why there is no spin polaron in 1D using the spin polaron language (magnons and holons). Moreover, due to the criticism raised by the Referee(s), we have decided to dig even deeper into the obtained result and to acquire quite a striking picture behind the spin polaron collapse [see new Sec. 5 and updated Sec. 1 and 7]. This can be summarised as following from the interplay of two mechanisms:
(1) The magnetic excitations are gapless in the 1D Heisenberg model due to the critical value of the magnon-magnon attraction in this model.
(2) The single hole experiences nonzero coupling to the gapless magnons: This is facilitated by an effectively fermionic nature of magnons in 1D that follows from their hard-core constraint.
These two conditions are special to 1D. This suggests a robust spin polaron in all t–J-like models that are not strictly 1D. Thus, the spin polaron quasiparticle is stable in all heavily investigated *quasi*-1D antiferromagnets such as SrCuO2, Sr2CuO3 or KCuF3. This suggests that any ARPES experiments on these compounds [below the Neel temperature] shows spin polaron physics, and the spin-charge separation is only approximate. Besides, the spin polaron might be a good quasiparticle in the quasi-2D doped antiferromagnets, even once the magnetic long range order collapses due to high hole doping. We leave the latter hypothesis as an important open problem for future studies.
We trust that the above detailed explanation of the obtained results, followed by a heavily revised version of the paper fully alleviates the Referee's doubts concerning this paper and that the manuscript now fulfils all the acceptance criteria of SciPost Physics.

---

## Round 4 · Referee Report · Anonymous (Referee 2) · 2023-11-6

Strengths
The authors find that the answer depends very sensitively on the spin symmetry of the parent spin Hamiltonian - the spin-polaron bound state is present in anisotropic XXZ models but disappears at the isotropic XXX point.
Weaknesses
It is unclear what new the present study adds to the old result. It seems that the differences between the Ising limit of the XXZ model and the isotropic XXX are well understood, both previously and also in the current manuscript, and are primarily due to the gapped nature of spin excitations. It is less clear what goes on in the XY-like (Luttinger) limit where spin excitations are gapless. Yet, the spin-polaron is present (Fig.3b). It would be nice to connect this limit with the previous extensive study of the problem in Ref.41 (which is not presently done).
Report
To conclude, I do not recommend the publication in SciPost unless significant revisions addressing the weaknesses listed above are made.
We are very grateful to the Referee for this very kind and extremely constructive report. Below we take the opportunity to address the questions raised by the Referee in great detail. In our opinion, these constructive comments, and the ensuing additional calculations and insights, have lead to a completely novel and thorough understanding of the spin polaron collapse. We believe that the heavily revised version of the paper fulfils all the acceptance criteria of SciPost Physics.
(A) "This is entirely consistent with previous studies of the question as described in Ref.33 and also in Phys. Rev. B 76, 115106 (2007). It is unclear what new the present study adds to the old result"
Indeed the most apparent explanation for the distinct behavior of the hole in the 1D t-J and 1D t-J^z model is that the magnetic excitations are gapless in the former and gapped in the latter case, cf. [Phys. Rev. Lett. 98, 266401 (2007)] and [Phys. Rev. B 76, 115106 (2007)]. For the purpose of this article, the above statement means that the spin polaron quasiparticle collapses in the 1D t-J model due to the onset of gapless excitations once the magnon-magnon interactions are tuned to their proper value in t-J model (and on the contrary, the spin polaron survives once the magnon-magnon interacitons are tuned away from this value).
While this statement is obviously *not* incorrect, we now present in the paper a far deeper understanding of the spin polaron collapse in the 1D AF [these additional calculations and the brought-by-them insight were heavily stimulated by another issue, see point (B) below, that had been raised by the Referee]:
(i) We start by showing that the mere onset of gapless excitations is not a sufficient condition for a quasiparticle collapse. While this statement follows, inter alia from general considerations on the fermion coupled to gapless bosons [see PNAS 111, 16314 (2014) and discussion below], we show on a specific example of a 1D holon-magnon model, with the holon coupled to a gapless magnons in a polaronic manner. The details of this consideration are given in the new Sec. 5.2 of the paper. The main point we want to make is that the collapse of quasiparticle in such systems crucially depends on the behavior of the magnon-holon vertex at such momentum transfers that the holon is coupled to a 0-energy manon: if this vertex vanishes, then the quasiparticle still survives, cf. new Fig. 9 of the paper.
(ii) Next, we show the origin of the spin polaron collapse in the 1D t-J^{XY} model (i.e. a model with solely XY interactions between spins, see also below):
Here the model is almost* exactly analytically solvable, since there are no interactions between magnons and the holon-magnon interaction in 1D can be well-accounted for by the self-consistent Born approximation (SCBA) which is exact in 1D. Crucially, we now show that the collapse of the spin polaron quasiparticle follow from the nonvanishing coupling between a holon and a gapless magnon in the 1D t-J^{XY} model. Moreover, the latter follows from the hard-core (or effectively "Jordan-Wigner" fermionic) nature of magnons in 1D -- since it is this statistics that leads to the fermionic Bogoliubov factors for magnons and to the nonzero magnon-holon coupling at vanishing energy-momentum transfers.
To sum up, we can say that the hard-core nature of gapless magnons in the 1D t-J^{XY} leads to the collapse of the spin polaron quasiparticle in this model. Note that this does *not* contradict the well-known paradigm that states that typically in electronic quasiparticles survive in system where electrons are coupled to (Goldstone) gapless bosons [cf. PNAS 111, 16314 (2014)]: the reason is that in the 1D t-J^{XY} model case we have gapless *hard-core* bosons, *not* Goldstone bosons.
[The above result is discussed in the new Sec. 5.3 (in particular, see Fig. 10).]
(iii) Finally, we discuss the impact of the results obtained for the 1D t–J^{XY} model on the understanding of the 1D t–J model. First, we note that the coupling between the fermionic holon and the hard-core bosonic magnon is not altered once the Ising interaction between spins is added to the 1D t–J^{XY} model. Hence, this coupling does not vanish at small energy-momentum transfers also in the 1D t–J model. Second, the magnetic excitations still remain gapless in the 1D t–J model. This is a fundamental property of the Heisenberg model. (In fact, the gapless excitations, and hence the conclusions that follow below, pertain to any 1D XXZ model with continuous sym-
metry of the spin interactions: i.e. of the 1D XY model, the easy-plane 1D XXZ model, and the 1D Heisenberg model.) Moreover, the magnetic excitations become gapped once the magnon-magnon interactions are fixed to their "proper" value givne by the 1D t-J model – as most easily visible by the onset of an effective staggered magnetic field once the magnon-magnon interaction are tuned away from that value. Intuitively, this is due to the fact that once the magnon-magnon attraction is not at this "critical" value, the valence-bond-like magnon ‘pair’ states are suppressed and the Neel-like states are favored.
Altogether, this means that the spin polaron quasiparticle collapses in the 1D t–J model is a result of the interplay of two effects: (i) onset of gapless magnon excitations at the (critical) value of the magnon-magnon
attraction that is inherent to the 1D t-J model, (ii) nonzero coupling between the fermionic holon and the gapless magnons at low-energy momentum-energy transfer that stems from the hard-core bosonic nature of the magnons.
[The above results is discussed in the end of the new Sec. 5.3 as well as in the revised Abstract, Introduction, abd Summary sections.]
Ad. * The model is not *exactly* solvable magnons are subject to their hard-core constraing. This means that once magnon-holon representation of the 1D t-J^{XY} model is later expressed in terms of the Jordan-Wigner fermions for magnons, there are the so-called string operators in the the magnon-holon coupling (see text ). These strings are neglected in the analytical solution but this does not seem to matter for the questiuon of the existence of the spin polaron quasiparticle (cf. the numerical solution in new Appendix C of the paper, which doest not show qualitatively different result w.r.t. the SCBA analytics).
(B) "It seems that the differences between the Ising limit of the XXZ model and the isotropic XXX are well understood, both previously and also in the current manuscript, and are primarily due to the gapped nature of spin excitations. It is less clear what goes on in the XY-like (Luttinger) limit where spin excitations are gapless. Yet, the spin-polaron is present (Fig.3b). It would be nice to connect this limit with the previous extensive study of the problem in Ref.41 (which is not presently done)."
We agree with the Referee that the Ising limit of the XXZ model and the isotropic limit are well understood, see also discussion above. Furthermore, we agree that indeed much less is known about the XY-limit (i.e. the t-J^{XY} model). In fact, Fig. 3b actually shows the XY model *but* in a staggered magnetic field [see Eqs. (28-29) with \lambda =1 in Appendix B) -- so this is not really the XY-limit. Besides, the mentioned by the Referee [Phys. Rev. B 57, 6444 (1998)] discusses this limit to some extent but not in great detail.
Taking the above into account, we have decided to first calculate the spectral function and ground state properties of the 1D t-J^{XY} model [see new Appendix C]. The numerically exact calculations unambiguously show that the spin polaron quasiparticle collapses (see new Figs 11-13). Next we fully connected this result to the one for the 1D t-J model -- and obtained crucial novel insights into the understanding of the spin polaron collapes in 1D t-J. We do not summarise these results here, since we already discussed them in great detail by answering to point (A) above.
(C) "The authors use their findings to question the standard interpretation of photoemission experiments on quasi-1D materials. They correctly point out that the staggered magnetic field produced by neighboring chains spoils the assumed SU(2) symmetry of a single spin chain and unavoidably stabilizes the spin-polaron. However, this appears to be an “academic” issue because the interchain staggered field appears only below the Neel transition temperature, which is very low for most of the relevant materials, which makes the critique irrelevant for actual experiments."
This is also a very interesting point. While we fully agree that the staggered field approach is stricly valid only below the Neel transition temperature, we still think that the critique discussed in this paper is relevant for actual experiments:
At higher temperatures, i.e. when there is no long-range order and the staggered field cannot be used to simulate the coupling between the chains, the (mentioned in the paper) fine balance between the magnon-magnon interaction and their on-site energies will also be disrupted due to the change of magnon on-site energies by the exchange interaction between the chains. Thus, only a specific ‘fluctuating-singlet’ (i.e. RVB-like) phase might preserve such a fine balance also in a quasi-1D system. While we are skeptical that this may indeed happen, we leave it as an important open problem for future verification using unbiased sophisticated numerical (this e.g. requires exact diagonalisation of a full 2D problem at finite temperature, which heavily suffers from finite size effects and is beyond the scope of this work, see also comment above.) Altogether, this means that the question of the survival of the spin-charge separation picture survives at T>TN in quasi-1D cuprates remains an open theoretical problem.
However, perhaps the more important insight is that the experimentally observed ARPES spectra at T>T_N do *not* constitute a proof of the onset of spin-charge separation in real materials neither at T>T_N nor at T<T_N. This is due to the following two facts:
First, as thoroughly discussed in this paper, the theoretical spectrum of the spin polaron in the quasi-1D antiferromagnets qualitatively resembles the experimentally observed ARPES spectrum of quasi-1D cuprates at T>T_N. In fact, it describes these spectra equally well as the spin-charge separation theory. At the same time, both theories suffer from a number of approximations (e.g., to the best of our knowledge, there are no realistic calculations showing the onset of spin-charge separation in coupled AF chains at T>T_N), when applied to the quasi-1D systems at T>T_N. Therefore, the so-far recorded ARPES spectra do *not* constitute a proof of the onset of spin-charge separation at T>T_N.
Second, at finite temperature, even for fermion-boson models with gapped bosons and hence a robust polaron at zero temperature, the quasiparticle peak broadens fast with increasing T and acquires all sorts of extra structure [M. Moeller & M. Berciu, PRB 90, 075145 (2014); J. Bonca et al., PRB 100, 094307 (2019)]. In other words, onset of "relatively" broad peaks in the ARPES spectra at finite temperature should not be regarded as a signature of the collapse of the quasiparticle picture at T<T_N.

---

## Editorial Decision

unknown